# A machine learning model trained on a high-throughput antibacterial screen increases the hit rate of drug discovery

A. S. M. Zisanur Rahman[1], Chengyou Liu[2], Hunter Sturm[3], Andrew M. Hogan[1], Rebecca Davis[3], Pingzhao Hu[2,4,5], Silvia T. Cardona[1,6]*

1 Department of Microbiology, University of Manitoba, Winnipeg, Manitoba, Canada, 2 Department of Electrical and Computer Engineering, University of Manitoba, Winnipeg, Manitoba, Canada, 3 Department of Chemistry, University of Manitoba, Winnipeg, Manitoba, Canada, 4 Department of Computer Science, University of Manitoba, Winnipeg, Manitoba, Canada, 5 Department of Biochemistry and Medical Genetics, University of Manitoba, Winnipeg, Manitoba, Canada, 6 Department of Medical Microbiology & Infectious Diseases, University of Manitoba, Winnipeg, Canada

* silvia.cardona@umanitoba.ca

**Data Availability Statement:** The machine learning model (D-MPNN) trained in this study was implemented in the open-source software Chemprop: https://github.com/chemprop/

## Abstract

Screening for novel antibacterial compounds in small molecule libraries has a low success rate. We applied machine learning (ML)-based virtual screening for antibacterial activity and evaluated its predictive power by experimental validation. We first binarized 29,537 compounds according to their growth inhibitory activity (hit rate 0.87%) against the antibiotic-resistant bacterium *Burkholderia cenocepacia* and described their molecular features with a directed-message passing neural network (D-MPNN). Then, we used the data to train an ML model that achieved a receiver operating characteristic (ROC) score of 0.823 on the test set. Finally, we predicted antibacterial activity in virtual libraries corresponding to 1,614 compounds from the Food and Drug Administration (FDA)-approved list and 224,205 natural products. Hit rates of 26% and 12%, respectively, were obtained when we tested the top-ranked predicted compounds for growth inhibitory activity against *B. cenocepacia*, which represents at least a 14-fold increase from the previous hit rate. In addition, more than 51% of the predicted antibacterial natural compounds inhibited ESKAPE pathogens showing that predictions expand beyond the organism-specific dataset to a broad range of bacteria. Overall, the developed ML approach can be used for compound prioritization before screening, increasing the typical hit rate of drug discovery.

## Author summary

Large-scale screening of chemical libraries can aid in the identification of bioactive lead compounds. However, the effectiveness of screening these libraries is impeded by associated expenses and limited chemical diversity. To address this problem, we trained a machine learning (ML) model with molecular properties of compounds that had antibacterial activity against the antibiotic-resistant bacterium *Burkholderia cenocepacia*. We then used the ML model to predict antibacterial activity in a library of FDA-approved

chemprop. The HTS dataset used for training, the trained weights, logs, Library compounds similarity comparisons, and results of both classification and regression models are available in the GitHub repository: https://github.com/cardonalab/Prediction-of-ATB-Activity.

**Funding:** STC, RD, and PH received a project grant No 169121 from the Canadian Institutes of Health Research (CIHR) https://cihr-irsc.gc.ca/, a basic research grant No 321858 from the Cystic Fibrosis Canada (CFC) https://www.cysticfibrosis.ca/, and a Pilot and Feasibility Grant No CARDON1810 from the Cystic Fibrosis Foundation (CFF) https://www.cff.org/. ASMZR was supported by a University of Manitoba Graduate Fellowship (UMGF) https://umanitoba.ca/ AMH was supported by a Vanier Canada Graduate Scholarship from the Government of Canada https://vanier.gc.ca/en/home-accueil.html HS was partially supported by the GETS program of the University of Manitoba https://umanitoba.ca/ The funders had no role in study design, data collection and analysis, decision to publish, or preparation of the manuscript.

**Competing interests:** The authors have declared that no competing interests exist.

compounds and an extensive, highly diverse virtual library of natural products, shortlisting compounds that could be further tested for activity in the laboratory. We confirmed the activity of top-ranked compounds against *B. cenocepacia* and ESKAPE pathogens. This work presents an ML approach that expands high-throughput screening datasets to the *in silico* identification of small molecules with potential activity and increases the hit rate of drug discovery.

## Introduction

Multidrug-resistant (MDR) bacterial infections present a serious threat to public health. In 2019, the Centers for Disease Control and Prevention (CDC) reported that approximately 3 million people suffer from MDR infections, resulting in about 35,000 deaths in the USA annually [1]. One of many multidisciplinary actions to address the antibiotic resistance crisis is the acceleration of early antibiotic discovery [2,3]. To achieve this goal, many programs have searched for antibacterial compounds in small molecule libraries using high-throughput screens (HTS). Target-based HTS searches for inhibitory compounds of a known target for which an *in vitro* activity assay is available [4]. These target-based approaches often fail to find active molecules for Gram-negative bacteria because most identified compounds do not penetrate the Gram-negative cell envelope [5]. Whole cell-based screens, where small molecule libraries are examined for inhibition of bacterial growth, can overcome this problem. However, the high costs and low success rate (1–2%) of HTS approaches have discouraged these efforts.

With the recent application of artificial intelligence (AI) in biology [6–8], merging drug discovery with AI has the potential to rapidly predict active molecules *in silico*, with a substantial decrease in associated expenses [9]. One key component of AI approaches in drug discovery is to obtain a computable representation of chemical molecules. In the field of machine learning (ML), convolutional neural networks (CNNs) can recognize these representations and detect patterns automatically, conducting convolutional operations [10]. On the other hand, graph convolutional networks (GCNs) apply the principles of convolution on chemical structures, represented as non-Euclidean structured graphs, in which nodes and edges of graphs represent atomic information (atomic number, formal charge, chirality, etc.) and bonding (bond type, conjugation, ring membership, etc.), respectively. Among the variants of GCNs, the directed-message passing neural network (D-MPNN) [11] successfully generates molecule-level representations by iterative message passing process on directed graphs. In brief, D-MPNN works by propagating atom and bond information in a directed manner during the message passing phase, resulting in a high-level feature (hidden state) for each atom in a molecule. In the readout phase, all hidden states of atoms are aggregated together and form a molecule-level feature vector, which can be fed into a feed-forward neural network (FFN) for the task-specific predictions.

The application of AI methods to antibiotic discovery is gaining attention [12]. Recently, deep learning was successfully used to discover antibiotic activity in compounds structurally unrelated to known antibiotics [13]. A library of approximately 2,300 structurally diverse molecules was compiled and evaluated for growth inhibition of *Escherichia coli*, and the binarized dataset was used to train a D-MPNN model to predict antibacterial activity in several virtual molecule libraries. The identified compounds displayed antibiotic activity against *Escherichia coli* and other bacterial pathogens. Here, we aimed to leverage this approach by applying it to an HTS against the Gram-negative bacterium, *Burkholderia cenocepacia*, previously performed

in our laboratory [14]. *B. cenocepacia* is part of the *Burkholderia cepacia* complex (Bcc), naturally antibiotic-resistant bacteria that cause infections in immunocompromised individuals [15]. Our goals were 1) to find new antibacterial compounds against *B. cenocepacia* and 2) to test the predictive power of the deep learning model for broad range activity when trained on HTS datasets performed in antibiotic-resistant bacteria. We first evaluated the predictive ability of our primary model on an FDA-approved compound library and found that the ML approach increased the hit rate to approximately 26%. Subsequently, we applied the trained model to a natural product library and identified a panel of growth inhibitory compounds active against both Gram-positive and Gram-negative pathogens, demonstrating that predictions of antibacterial activity against *B. cenocepacia* could be partially extrapolated to other bacterial pathogens. Two active compounds, STL558147 and PHAR261659, had no previous record of antibacterial activity. STL558147 is structurally similar to rifampicin and has broad-spectrum activity, including the ESKAPE pathogens [16]. The structure of PHAR2611659 does not resemble any compound with a previous record of antibiotic activity. Our work extends the applicability of previously developed whole-cell HTS to the training of ML models, increasing the hit rate of subsequent screening campaigns.

## Results

### Training a deep learning model with a *B. cenocepacia* HTS Dataset

To train our deep learning model, we repurposed our previously performed HTS that searched for growth inhibitory compounds against *Burkholderia cenocepacia* strain K56-2 [14]. The dataset used in the ML approach consisted of 29,537 compounds with residual growth (RG) values and average B-scores [17]. The RG measures the ratio of bacterial growth in the presence and absence of the compounds. The B-score measures relative potency that adjusts the RG for any screening artifacts resulting from well position (row and column) in the assay plate during the HTS. The B-score is inversely proportional to compound potency, where negative B-scores indicate greater growth inhibitory activity of the compounds. To binarize the compounds, the previously established average B-score threshold of -17.5 was chosen [14]. Overall, 256 small molecules were classified as growth inhibitory (**Fig 1A**).

Next, the dataset was applied to a directed-message passing neural network (D-MPNN) approach [11] to model the training data and make predictions. The D-MPNN was previously used to train a binary classification deep learning model with 2,335 compounds for predicting antibiotic activity [13]. We reasoned that using the D-MPNN training with a larger library (approximately 30,000 compounds) would enhance the model's ability to learn, generalize and make predictions across compound libraries outside of the training dataset. To maximize the utilization of inactive compounds, we did not enforce class balance while training. The trained model generated scores between 0 and 1 for each molecule, with 1 indicating the highest probability for growth inhibitory activity. In addition to the binary classification, we also conducted regression tasks on both the average B-score and RG, where the two potency measurements of activity against *B. cenocepacia* were used simultaneously to train multi-task models.

To enhance the accuracy of the predictions, the model was additionally supplemented with molecular fingerprints or descriptors in the first layer of the FFN (**Fig 1B**). We also applied two standard machine learning optimization strategies to increase the robustness of the models: Bayesian hyperparameter optimization [18] and ensembling [19]. The iteration of Bayesian optimization was set to 20 iterations, and the ensemble size was equal to 5 for each model. Four models with different combinations of molecule-level features (**Table 1**) were trained and evaluated for classification and regression tasks. In addition, two different splitting strategies were used to build the ML models, a scaffold split and a random split (**S1**–**S4 Tables**). The

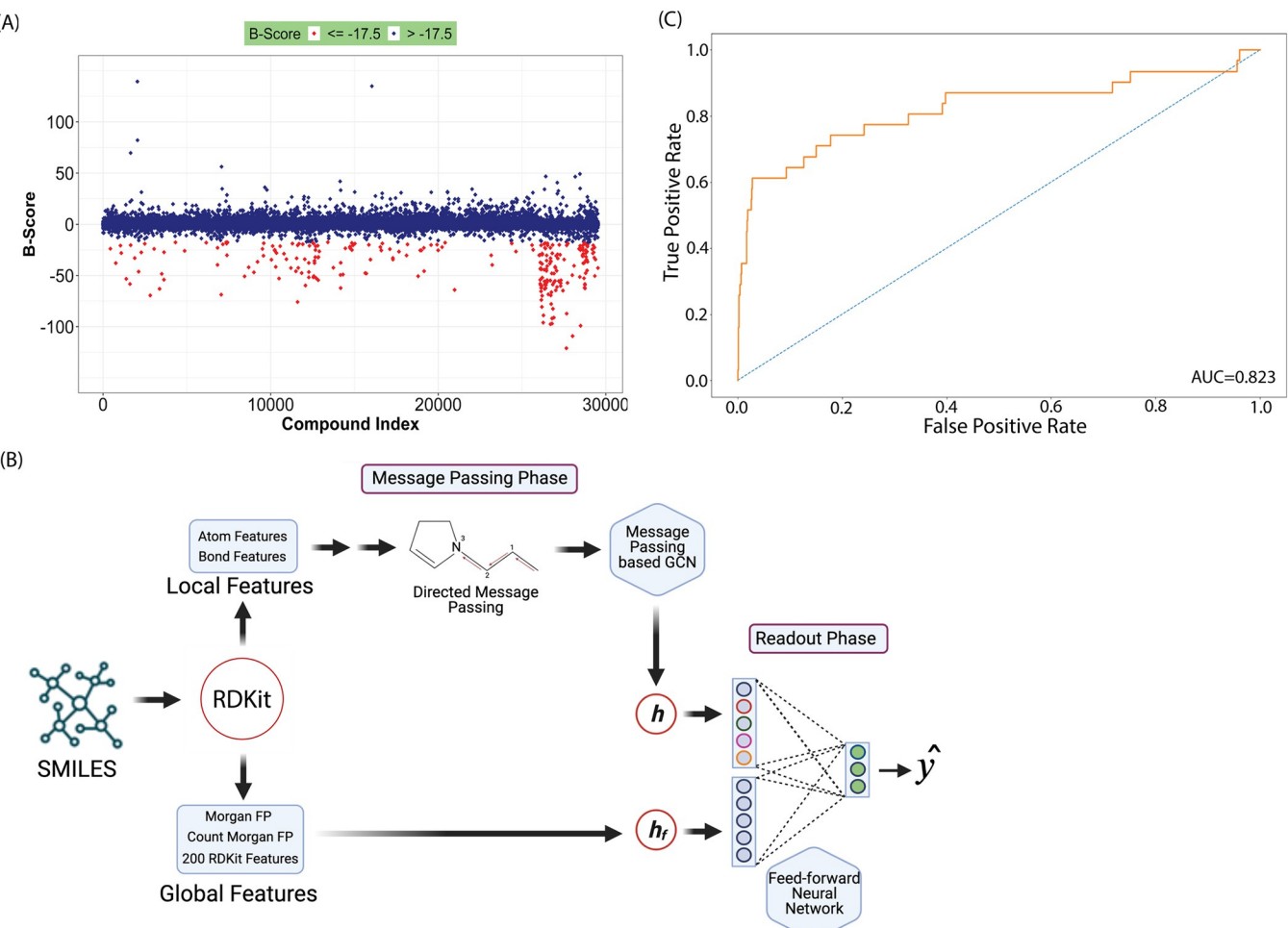

**Fig 1. Initial training and performance evaluation of the machine learning model.** (A) High-throughput screening data generated by screening a compound library of 29,537 compounds against *B. cenocepacia* K56-2 wild-type. Using B-score ≤ -17.5 as a threshold, the screening yielded 256 active compounds. Dark blue and red represent inactive and active compounds, respectively. (B) The machine learning model was trained using a D-MPNN approach, which extracts compounds' local features, such as atom and bond features. The model was fed more than 200 additional global molecular descriptors to further increase the accuracy. Dataset was split into 80:10:10 ratio to train, validate and test the model. (C) ROC-AUC plot evaluating model performance after training. The model attained a ROC-AUC of 0.823. Parts of panel B are modified from Yang *et al.* [11]. Fig 1B was created with https://biorender.com/.

scaffold split separates samples into subsets based on molecular scaffolds. From the eight different combinations, Model 6 (binary classification, scaffold split trained with RDKit descriptors, S2 Table) achieved the highest area under the curve of the precision-recall curve (PRC-AUC = 0.241), F1 Score (F1 = 0.104), Matthews correlation coefficient (MCC = 0.167) on the test set and was therefore selected as the primary model for our subsequent

**Table 1. Deep learning models trained with different combinations of molecule-level features.**

| Additional Features | Description |
|---|---|
| Combination 1 | D-MPNN without molecule-level features |
| Combination 2 | D-MPNN with RDKit descriptors |
| Combination 3 | D-MPNN with RDKit descriptors and count-based Morgan fingerprints |
| Combination 4 | D-MPNN with RDKit descriptors and binary Morgan fingerprints |

experiments. Model 6 also attained the second-highest ROC-AUC score (0.823) among other baselines in the same category (**Fig 1C**).

## Predicting and confirming growth inhibitory activity in virtual compound libraries

To validate the model's ability to make predictions on compounds outside of the training dataset, we employed Model 6 on an FDA-approved compound library. To ensure that no identical compounds were present in both the training set and the FDA-approved library, we compared both libraries and compounds with identical SMILES strings were filtered out from the FDA-approved library. We then predicted the growth inhibitory activity of 1,614 compounds that were not present in the training set (**S5 Table**). The model generated a single value between 0 and 1 for each molecule, indicating the probability of the molecule being active. The 100 top-ranked compounds contained a large fraction of antibiotics (49%), antimicrobials, and antineoplastic agents. (**S1 Fig**). After removal of duplicated compounds, we tested 81 commercially available compounds for growth inhibitory activity (at least 20% of normal growth inhibition) against *B. cenocepacia* K56-2 (**Fig 2A** and **S6 Table**). Therefore, our threshold of residual growth (RG) was 0.8. Although Selin et al. used an RG of 0.7 [14], we reasoned that establishing a permissive RG threshold would allow us to capture compounds with potential antimicrobial activity that could be further optimized for potency. Twenty-one compounds were experimentally validated as growth-inhibitory, establishing a positive predictive value (PPV) of 26% (**S6 Table** and Figs **2B** and **S2**). Overall, 14 of the 21 compounds exhibiting growth inhibitory activity were known antibiotics or antimicrobial compounds (**Fig 2C**). A correlation (Pearson correlation test, R = 0.54) was found between bioactivity and the predicted rank. Compounds that ranked higher based on the prediction score displayed stronger growth inhibitory activity (**Fig 2D**).

To extend the applications of our model, we predicted growth inhibitory activity in a natural product virtual library containing 224,205 compounds (**S7 Table**). We ranked the compounds based on their predicted score and filtered out those with previously reported antimicrobial activity or toxicity. From the 100 top-ranked unique compounds, 43 compounds were tested against *B. cenocepacia* K56-2 (**S8 Table**). The screening yielded five previously uncharacterized small molecules with growth inhibitory activity against *B. cenocepacia* K56-2, achieving a PPV of 12% (**Fig 3A**). Compared to the hit rate from our previously performed whole-cell-based high-throughput screens (0.87%) [20], our strategy improved the hit rate at least 14-fold, suggesting that our *in silico* approach can substantially minimize costs and time associated with compound screening.

To examine the predictive power of the deep learning model for a broad-range antibacterial activity, we tested the 43 curated compounds from the natural product library against five ESKAPE pathogens- *Acinetobacter baumannii*, *Enterobacter cloacae*, Methicillin-resistant *Staphylococcus aureus* (MRSA), *Klebsiella pneumoniae* and *Pseudomonas aeruginosa*. The screening identified 22 compounds with growth inhibitory activity against at least one of the species tested (PPV = 51.16%; **Fig 3B**). While 13 compounds were active against Gram-positive MRSA ATCC33592, three, nine, five, and nine were active against *E. cloacae* ENT001_EB001, *P. aeruginosa* PAO1, *K. pneumoniae* ESBL_120310 and *A. baumannii* 1225, respectively (**Fig 3B**). Among the inhibitory compounds identified from the screening, STK760075 (ZINC000004524372), PHAR261659 (ZINC000008876405), 0167–0032 (ZINC000000254599), 6414–0936 (ZINC000000465709) and STL558147 (ZINC001286671837) displayed the widest range of activity against the pathogens tested (**Fig 3**). To the best of our knowledge, none of these compounds have previously been reported to

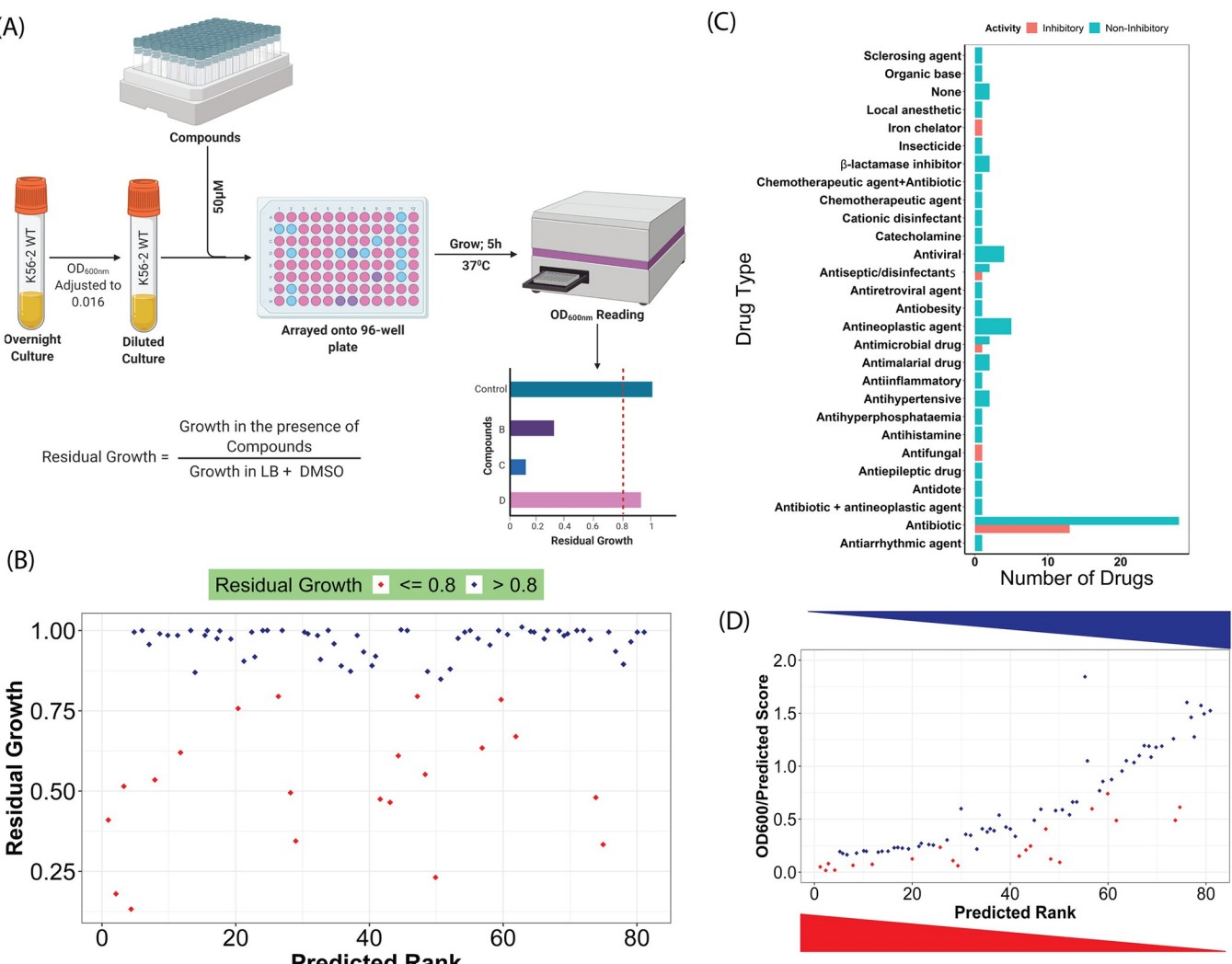

**Fig 2. *In vitro* testing of top-ranked predicted compounds from an FDA-approved compound library.** (A) Schematic of the screening protocol. Eighty-one commercially available compounds (from the top 100) were screened. (B) The screening identified 21 bioactive compounds with a positive predictive value (PPV) of 25.9%. Dark blue and red represent inactive and active compounds, respectively. (C) The top 100 ranked compounds selected for empirical testing belong to different drug families. Most of the compounds exhibiting bioactivity were known antibiotics or antimicrobial compounds. (D) The ratio of OD$_{600nm}$ and prediction scores were plotted against the predicted rank of the corresponding compounds. The results show a linear correlation (Pearson correlation, R = 0.54) between the prediction score and bioactivity. The predicted score is the probability of a compound being active as calculated by the ML model. The predicted rank is the order of the compounds based on the predicted score, where compounds with the higher predicted scores are ranked higher. The red and blue triangles show the gradient of predicted rank and growth (measured as OD$_{600nm}$), respectively. Dark blue and red indicate compounds' probability of being inactive and active, respectively. Results are the average of at least three independent biological replicates. Fig 2A was created with https://biorender.com/.

have growth inhibitory activity. STK760075 exhibited activity against *B. cenocepacia* K56-2, MRSA ATCC33592, and *P. aeruginosa* PAO1, whereas PHAR261659 was active against *E. cloacae* ENT001_EB001, *P. aeruginosa* PAO1, *K. pneumoniae* ESBL_120310 and *A. baumannii* 1225 (**Fig 3**). Compound 6414–0936 exerted inhibitory activity against four of the six pathogens tested (except *B. cenocepacia* K56-2 and *E. cloacae* ENT001_EB001). Compound 0167–0032 had inhibitory activity against all pathogens tested, except *K. pneumoniae* ESBL_120310. Compound STL558147 was active against all six species tested. While 0167–0032 (PubChem CID 1560156) and STK760075 (PubChem CID 1549520) are annotated as irritants in the PubChem database, no known toxicity data is available for other compounds with broad-range

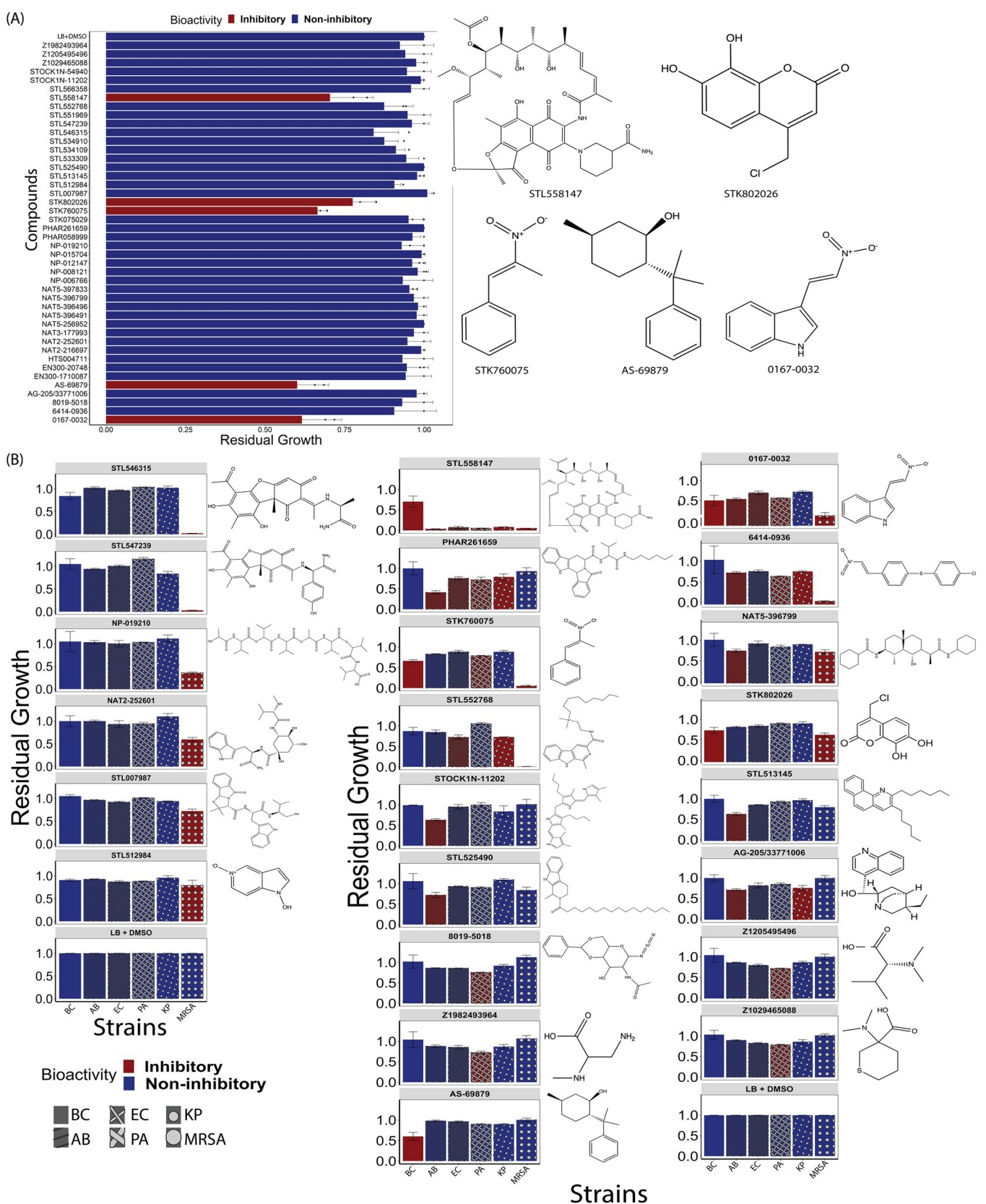

**Fig 3. *In vitro* testing of top-ranked predicted compounds from an unprecedented natural product library.** (A) The 43 commercially available compounds (from the 100 top ranked unique compounds) were screened against *B. cenocepacia* K56-2. The screening yielded 5 bioactive compounds with a hit rate 10 times higher than a conventional screening (hit rate = 11.63%). Dark blue and red are non-inhibitory and inhibitory compounds, respectively, based on the residual growth (RG) threshold of 0.8. (B) Screening these 43 compounds against the ESKAPE pathogens yielded 22 bioactive compounds that displayed broad-spectrum growth inhibitory activity against diverse pathogens (Positive predictive value (PPV) = 51.16%). Dark blue and red are non-inhibitory and inhibitory compounds, respectively. The structures of the compounds that exhibited growth inhibitory activity against *B. cenocepacia* K56-2 and the ESKAPE pathogens are shown beside the plots. Results are average of at least three independent biological replicates. Error bars indicate mean ± SD. AB = *A. baumannii* 1225, BC = *B. cenocepacia* K56-2, EC = *E. cloacae* ENT001_EB001, PA = *P. aeruginosa* PAO1, MRSA = Methicillin-resistant *S. aureus* ATCC33592.

activity. For this study, we selected STL558147 and PHAR261659 for further characterization since they exhibited the strongest growth inhibitory activity against most of the pathogens tested (**Fig 3**).

## Antibacterial properties of STL558147

To confirm the identity of STL558147, we performed nuclear magnetic resonance (NMR) spectroscopy and compared the spectrum with that of STL558147 provided by the supplier (**S3 Fig**). We found that STL558147 and the antibiotic rifampicin share a similar scaffold (Maximum common substructure Tanimoto score 0.6389) but are different compounds (**Figs 4A and** S3).

Rifampicin targets the β subunit of the bacterial DNA-dependent RNA polymerase [21,22], encoded by *rpoB*. To test if STL558147 targets RpoB, we explored the link between RpoB expression and susceptibility to STL558147, as target depletion often sensitizes cells to cognate antimicrobials [23]. We created a knockdown mutant of the *rpoBC* operon in *B. cenocepacia* K56-2 (K562_RS01210-5) using CRISPR interference (CRISPRi) [24]. The CRISPRi system developed for *Burkholderia* comprises a chromosomally integrated *dCas9* from *Streptococcus pyogenes* placed under the control of an L-rhamnose-inducible promoter and plasmid-borne target-specific single guide RNA (sgRNA) driven by a constitutively active synthetic promoter $P_{J23119}$ [25]. Addition of rhamnose induces *dCas9* expression, which binds the sgRNA and sterically blocks transcription of the target specified by the sgRNA. *B. cenocepacia* K56-2 *gyrB* CRISPRi knockdown mutant (K562_RS02180) was included as a negative control. When grown at the rhamnose concentration that inhibits 50% of growth (Rha $IC_{50}$) compared to wild-type, the *gyrB* CRISPRi mutant was more susceptible to novobiocin than K56-2 WT and non-targeting controls (**Fig 4B**). This was expected as GyrB is the target of novobiocin [26]. Similarly, the *rpoBC* knockdown mutant exhibited enhanced sensitivity against both rifampicin (**Fig 4C**) and STL558147 (**Fig 4D**). The hypersensitivity of the *rpoBC* CRISPRi mutant against both rifampicin and STL55817 suggests that STL558147 has a similar mechanism of action to rifampicin, and RpoB is likely the target of STL558147. Rifampicin is known to have synergistic interactions with colistin and meropenem *in vitro* against *Pseudomonas* spp., *Acinetobacter* spp., and carbapenemase-producing *Enterobacteriaceae* [27–30]. To elucidate if STL558147 exerts synergistic interactions with clinically relevant antibiotics against *B. cenocepacia* K56-2, we performed a microdilution checkerboard assay. Using the Bliss interaction and the Loewe additivity scores, we considered scores >15 as synergistic and < -15 as antagonistic [31]. We observed a strong synergistic interaction when STL558147 was combined with either ceftazidime, colistin or polymyxin B (**Figs 5A and** S4A **and S9 Table**), similar to the synergistic interaction observed between ceftazidime and avibactam (**Fig 5B**). Compared to rifampicin, the observed synergistic activity of STL558147 was stronger with ceftazidime, colistin and polymyxin B (**Fig 5**). The observed synergistic activity of rifampicin was weaker than the commonly used drug pair ceftazidime and avibactam (**Fig 5**). When STL558147 was combined with these antibiotics against *B. cenocepacia* K56-2, we observed similar growth

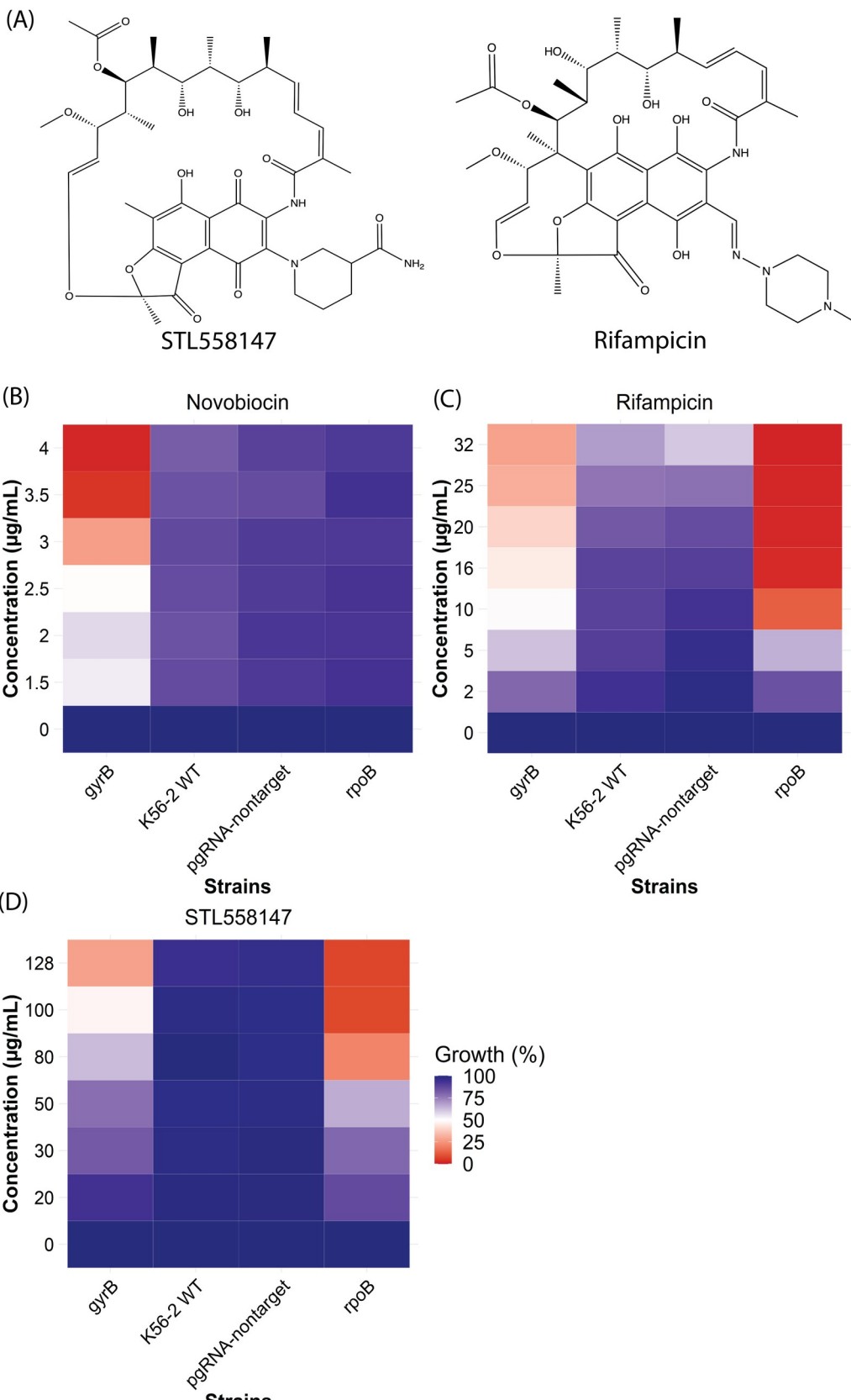

**Fig 4. Enhanced sensitivity of the CRISPRi knockdown mutants indicated RpoB as the *in vivo* target of STL558147.** (A) Chemical structures of STL558147 and Rifampicin. (B-D) Comparison of hypersensitive CRISPRi knockdown mutants to novobiocin (B), rifampicin (C) and STL558147 (D). Blue indicates more growth (less inhibition), and red indicates less growth (more inhibition). Results are average of at least three independent biological replicates.

inhibitory activity at a lower STL558147 concentration compared to STL558147 alone (**Figs 5A** and S4A). We observed no interaction between STL558147 and rifampicin or rifabutin (S5A **and** S5B **Fig and** S9 Table). This is expected as both rifampicin and rifabutin target RpoB, which is also the likely target of STL558147. On the contrary, the combination of STL559147 with ciprofloxacin was antagonistic (**S5C Fig and** S9 Table).

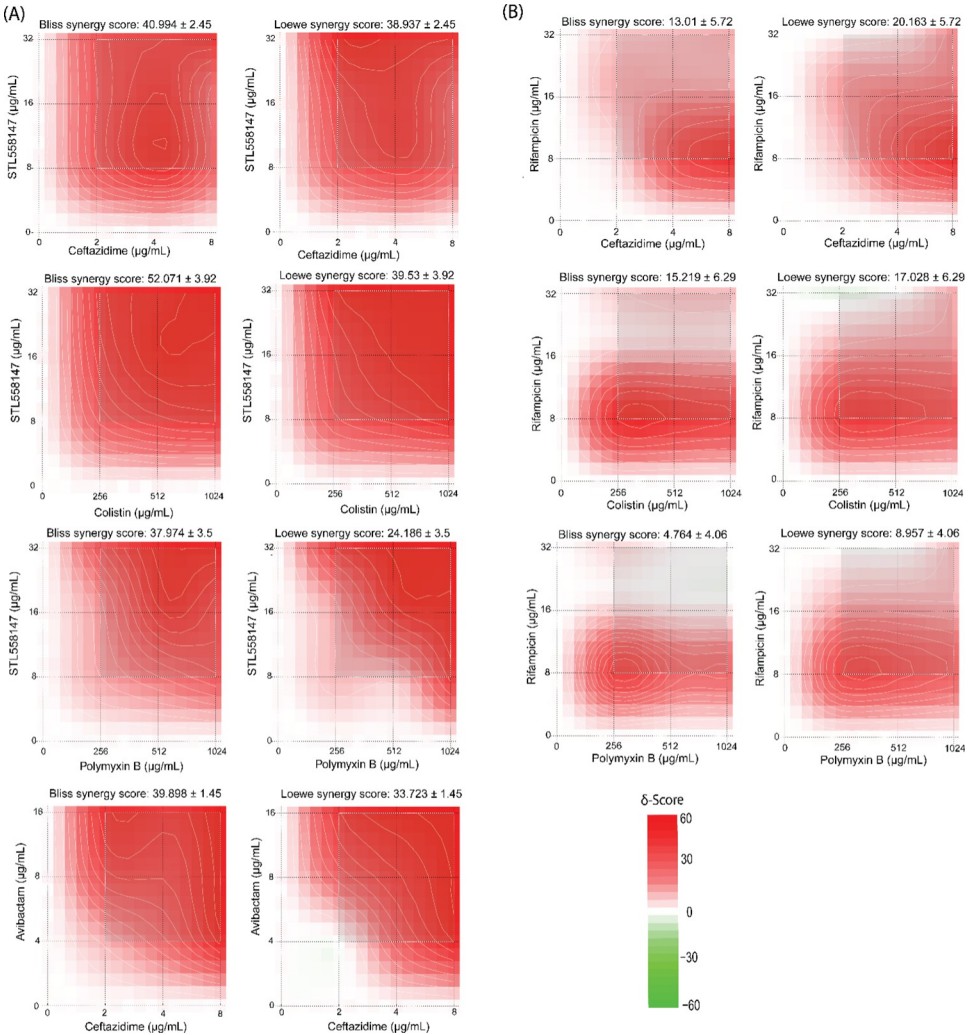

**Fig 5. Synergy maps of STL558147 and rifampicin combined with other antibiotics against *B. cenocepacia* K56-2.** Synergy plots of STL558147 (A) and rifampicin (B) with ceftazidime, colistin, and polymyxin B. The synergy scores were calculated based on the widely used Bliss independence [52] and Loewe additivity [53] models. The most synergistic area in each combination is highlighted with a rectangular box inside the plot. Green (negative δ-scores) indicate antagonistic interactions, and red (positive δ-scores) indicate synergistic interactions. Synergy scores >15, between -5 to 15, and < -15 were considered synergistic, additive and antagonistic, respectively. Results are average of at least three independent biological replicates. Synergy scores are shown as mean ± SEM. Synergy scores were calculated using SynergyFinder 2.0 [31].

### PHAR261659 Exhibited broad-spectrum antibacterial activity

Screening of the 43 compounds from the 100 top-ranked unique natural products (**S8 Table**) revealed that PHAR261659, a compound with no previously reported antibiotic activity, has antibacterial activity against a broad range of pathogens (**Fig 3B**). Specifically, PHAR261659 exhibited growth inhibitory activity against *A. baumannii* 1225, *E. cloacae* ENT001_EB001 and *K. pneumoniae* ESBL_120310 (**Fig 3B**). However, we did not observe growth inhibitory activity against *B. cenocepacia* K56-2 and *S. aureus* ATCC33592 (MRSA). Moreover, while PHAR261659 displayed bioactivity at the screening concentration (50μM), the compound was not soluble beyond 64μM. However, we found 15 unique analogs with modified side chains and lower *logP* values in the natural product library (**S10 Table**) and tested them against *B. cenocepaci*a K56-2 and the *ESKAPE* pathogens. While PHAR261659 did not show bioactivity against *B. cenocepacia* K56-2 and methicillin-resistant *S. aureus* ATCC33592 (**Fig 3**), five and ten analogs exhibited growth inhibitory activity against these pathogens (**Fig 6**). Particularly, STL529920 (ZINC000008876407), a stereoisomer of PHAR261659 (**S6 Fig**), was active against all six pathogens tested (**Fig 6**). Both STL529920 and PHAR261659 were predicted to be active by our deep learning algorithm with scores of 0.48 and 0.49, respectively (**S7 Table**).

## Discussion

The application of artificial intelligence in biomedical sciences has shown promising outcomes, from diagnosis [32] to disease prediction from medical records [33] to treatment response [34]. Using high-throughput compound screening datasets to train ML models for drug discovery represents another application of machine learning in the biomedical science field. Specifically, the application of machine learning to antibiotic discovery has demonstrated promise as a next-generation strategy to address the crisis of antibiotic resistance [13,35,36].

Here, we demonstrate that ML approaches that use algorithmic solutions to identify novel structural classes of antibiotics can decrease the associated cost and time of HTS by allowing *in silico* exploration of vast, diverse chemical spaces that are otherwise unprocurable [35,37]. In the present study, we applied a machine learning model using a directed-message passing neural network (D-MPNN) that learns properties of compounds by sending messages along the atoms in a directed fashion [11]. We trained the algorithm with a high-throughput screening dataset performed against *B. cenocepacia* K56-2 along with >200 computationally extracted compound features [14,38]. After evaluating the trained model in an FDA-approved library (**Fig 2**), we subsequently applied it to an unprecedented natural product library containing over 200,000 compounds and identified multiple growth inhibitory compounds (**Fig 3**). A similar strategy was applied to identify an antibiotic with a novel mechanism of action [13]. We observed an improvement of 14-fold compared with the previously obtained hit rate. Our hit rates were lower than those reported by the Collins group [13]. However, the small molecule library used in our HT screening was not previously curated for its use in training ML models. A training dataset with random chemical diversity and an imbalance between the number of active and inactive compounds may have contributed to the lower hit rate observed and reduced the model's ability to generalize to new scaffolds. Indeed, the predictive power of machine learning algorithms is highly susceptible to datasets with limited diversity [39].

Also, the experimental systems and the established threshold that define the hits can impact the hit rates of HTS screening campaigns and hence the binarization of the training sets. Nevertheless, we expect any robust HTS dataset can be used for machine learning-based predictions of biological activity. This is a hypothesis currently being tested by our group.

Before predicting antimicrobial activity in the FDA-approved library, we filtered out the duplicate compounds between this library and the HTS dataset. However, upon estimating

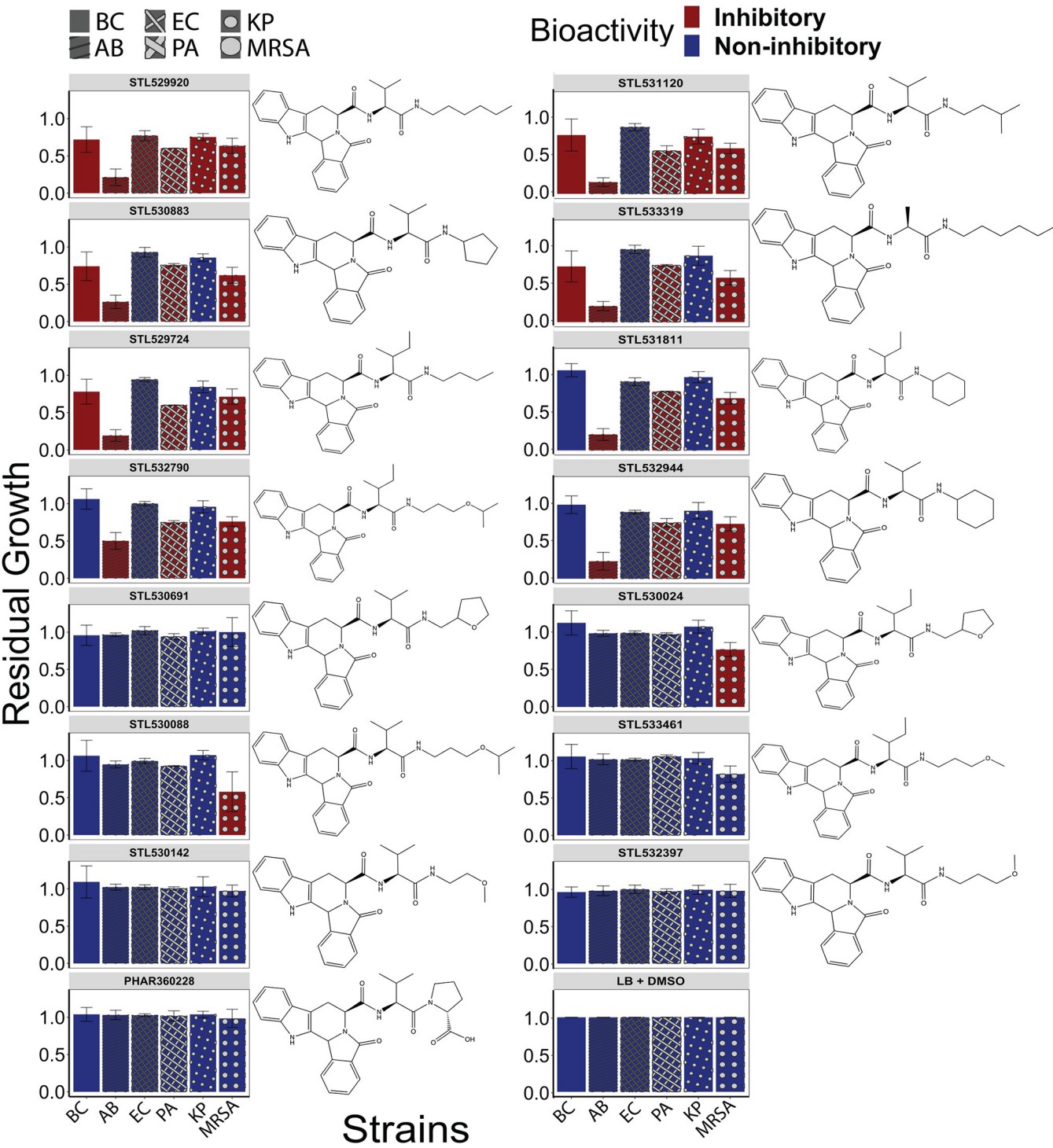

**Fig 6. Screening of PHAR261659 analogs.** PHAR261659 analogs with different side chains were selected based on lower predicted *logP* values. STL529920, a stereoisomer of PHAR261659, exhibited growth inhibitory activity against all six pathogens tested. The activity of growth inhibitory and non-growth inhibitory compounds are shown in red and blue, respectively. Results are the average of three independent biological replicates. Error bars indicate mean ± SD.

similarity by the Tanimoto Combo (TC) score, two compounds present in the HTS dataset (clioquinol and cetylpyridinium) had identical duplicates in the FDA-approved library (**S7A Fig**). None of these compounds was predicted to have growth inhibitory activity. The next most similar compounds between the two libraries received had TC scores between 1.95 and 1.63 (**S7B Fig**).

Our study identified multiple compounds with broad-range growth inhibitory activity (**Fig 3**). Compounds 0167–0032 (PubChem CID 1560156), PHAR261659 (no known PubChem ID) and STK760075 (PubChem CID 1549520) inhibited the growth of five, four and three pathogens, respectively, from the pathogens tested (**Fig 3B**). Although 0167–0032 and STK760075 are marked as irritants in the PubChem database, some clinically used antibiotics are also known to cause discomfort at higher concentrations. For example, clinically used common antibiotics meropenem (PubChem CID 441130), penicillin G (PubChem CID 22502), cephalexin (PubChem CID 27447), and ampicillin (PubChem CID 6249) are also labelled as irritants or allergenic by GHS (Globally Harmonized System of Classification and Labelling of Chemicals). While our study identified multiple bioactive compounds (**Fig 3**), STL558147 was the most potent and had broad-spectrum activity (**Fig 3B**). STL558147 (ZINC001286671837) obtained a high probability score of being growth-inhibitory (0.833; **S8 Table**), probably due to its structural similarity to rifampicin (**Fig 4A**) which was present in the training dataset [14]. We are unsure about the origin of STL558147, but the compound is likely synthesized by *Amycolatopsis sp.* (with unknown isolation source), similar to rifampicin.

With the rise of antimicrobial resistance, combinatorial antibiotic treatments can be an effective therapeutic strategy to prevent antimicrobial resistance. Combinatorial antibiotic strategies can achieve a therapeutic effect at relatively lower concentrations, decreasing adverse and toxic effects and severely restricting the acquisition of drug resistance. Rifampicin is known to have synergistic effects against a range of Gram-negative pathogens when used in combination with other antibiotics [27–30]. We observed a synergistic growth inhibitory effect of rifampicin with ceftazidime and colistin (no interactions with polymyxin B) (**Figs 5B and** S4A). Like rifampicin, STL558147 also demonstrated synergistic interactions with ceftazidime and colistin *in vitro* (**Figs 5A and** S4A). This was expected since STL558147 is structurally very similar to rifampicin and likely has the same mechanism of action (**Fig 4**). While rifampicin did not display any synergistic interaction with polymyxin B, we observed a strong synergy between STL558147 and polymyxin B *in vitro* (**Figs 5 and** S4). Colistin and polymyxin B disrupts the bacterial outer membrane [40], whereas ceftazidime interacts with penicillin-binding protein 3 (PBP3) to inhibit cell wall synthesis [41]. Probably, this weaker outer barrier allows more STL558147 influx into the cell, resulting in greater growth inhibition at lower STL558147 concentration. STL558147 has a relatively high minimum inhibitory concentration (MIC) (256μg/mL) compared to rifampicin (64μg/mL) against *B. cenocepacia* K56-2. However, STL558147 exerted stronger synergistic growth inhibitory activity with ceftazidime, colistin and polymyxin B at the same drug concentrations (**Figs 5 and** S4), warranting the further development of STL558147 to generate a more potent derivative.

While PHAR261659 exhibited a broad range of bioactivity, the compound was not active against *B. cenocepacia* K56-2 and methicillin-resistant *S. aureus* ATCC33592. Screening for analogs of PHAR261659, we found STL529920, a stereoisomer of PHAR261659, to be active against all six pathogens tested. This finding is not surprising as stereoisomers often differ in their biological activity primarily due to the stereoselectivity of the drug-target interactions [42]. Such interactions may have steric constraints due to the three-dimensional spatial arrangement of the functional groups within the drug molecule and may alter the functionality and efficacy. For example, DR-3355, the S isomer of ofloxacin, was found to be two times

more potent than the S isomer primarily due to stronger binding interactions of the S isomer with its target, DNA gyrase [43].

Overall, our work demonstrates that training a deep learning model with previously developed screening datasets can increase the hit rate of such screening efforts. While we have demonstrated the utility of our model for antibiotic drug discovery, we believe that our approach can be expanded to other HT screens, increasing the repertoire of potentially bioactive molecules.

## Conclusions

We utilized an antibacterial HTS dataset previously performed against an antibiotic-resistant bacterium to train a machine learning model in the discovery of antibacterial molecules of a broad-range spectrum. Bioactivity predictions in virtual FDA-approved and natural product libraries identified novel compounds with experimentally confirmed growth inhibitory activity against several Gram-negative bacteria. Our approach increases the hit rate of the previous drug discovery effort by at least 14-fold.

## Methods

### Deep learning model details

The code of D-MPNN used in this study is implemented in the package Chemprop [44], which was built based on the architecture proposed by Gilmer *et al.*, named message passing neural network (MPNN) [45]. In general, MPNNs take atom and bond features as inputs and aggregate the features through a message-passing phase and a readout phase [45]. Different from the message passing mechanism of MPNN, in which molecular messages are centered on atoms, D-MPNN propagates molecular information through neighbouring bonds with directions. As D-MPNN aggregates messages associated with directed bonds, it can avoid unnecessary loops and totters. This property allows D-MPNN to become more efficient during training and is able to construct a more informative featurization with less noise. The readout phase of D-MPNN follows the same paradigm of typical MPNNs, in which hidden states for atoms in a molecule are aggregated together and form a molecule-level representation.

### Supplementary molecular features

Due to the limited ability of extracting molecule-level features in the message passing phase, especially in the case of large molecules, we choose to supplement the learned graphical representations from D-MPNN with calculated molecule-level features. In this study, we investigated three types of molecular features as additional auxiliary to D-MPNN architecture: binary Morgan fingerprints, count-based Morgan fingerprints, and 200 chemical descriptors extracted with the chemoinformatics software RDKit [38]. To avoid the effect of large range additional features dominating other features, normalization was applied to the additional features to scale the values to a fixed range before entering into networks. In particular, min-max scaling was applied to normalize count-based Morgan fingerprints, and 200 RDKit descriptors were normalized by fitting to the cumulative density functions.

### Bayesian optimization

To maximize the performance, we performed Bayesian hyperparameter optimization using the Hyperopt package [46]. We optimized four hyperparameters in D-MPNN: the number of the message passing steps, the hidden size of the neural network, the number of FFN layers, and dropout probability. The Bayesian hyperparameter optimization uses the results of prior

trials to make informed decisions for what parameter values to try in the subsequent trial. Compared to the grid search method, this method requires fewer iterations to find the optimal hyperparameters.

### Ensembling

Ensembling, a commonly used technique in machine learning for improving a model's performance, was also applied in our training process. The ensemble method combines the predicted results from several identically structured models with different initial weights. That is, models are trained independently and separately, then the prediction values were averaged with equal weights, resulting in the final prediction [19].

### Deep learning model training and prediction

The deep learning model was trained with an HTS dataset that used the Canadian Compound Collection (CCC) against *B. cenocepacia* K56-2 [14]. The training dataset consisted of 29,537 compounds. The textual molecular representation from the SMILES strings was transformed into numeric representations to generate molecular features during training. Similar to Selin *et al.*, [14], we used Residual Growth (RG) ($< 0.7$ RG) and average B-Score ($< -17.5$) to classify compounds as growth inhibitory activity against *B. cenocepacia* K56-2.

For the classification task, we used the average B-Score -17.5 as the bioactivity threshold. Compounds with an average B-Score of less than -17.5 were considered growth inhibitory, resulting in 256 active compounds. Although the data is severely imbalanced, we decided against enforcing class balance on data during training to maximize the utilization of data. With the determined threshold, the active compounds were labelled as 1 while the rest were labelled as 0, then used as targets for classification training. After training, the D-MPNN output was a single value between 0 and 1 for each molecule, indicating the compound's probability of having growth inhibitory activity. For the regression task of D-MPNN, both the average B-Score and RG were used in the training together but evaluated separately in the results.

### Data splits

The data were divided into training, validation, and testing sets following the ratio 80:10:10 with the data split strategy named scaffold split. Unlike the random split commonly used in ML, the scaffold split is designed explicitly for the QSPR/QSAR tasks, where the Murcko scaffold for each molecule is calculated and used during the splitting process [47]. Assigning compounds to data bins based on Scaffold scores enforces validation and testing sets to have more molecular diversities and reduces similarities between data sets. Therefore, the scaffold split strategy has a more realistic and challenging evaluation compared to the random split [11]. Besides the scaffold split, we also trained models on data bins randomly partitioned as a comparison.

### Evaluation methods

After binarizing the results, we evaluated our binary classification models by four metrics: ROC-AUC, PRC-AUC, F1 score, and Matthews correlation coefficient (MCC) [48,49]. The cut-off of binarization for each model was determined by the Youden's J statistic on the ROC curve of the validation set, which is defined as:

$$J = sensitivity + specificity - 1$$

After obtaining optimum cut-offs, we computed F1 scores and MCC to further evaluate the performance of classifiers. F1 Scores and MCC were calculated using the following formulas:

$$F1\ Score = 2 \times \frac{precision \times recall}{precision + recall}$$

$$MCC = \frac{TP \times TN - FP \times FN}{\sqrt{(TP + FP)(TP + FN)(TN + FP)(TN + FN)}}$$

where TP, TN, FP, FN respectively represent the number of true positives, true negatives, false positives, and false negatives.

For regression tasks, we used RMSE and MAE scores as measures of the differences between the ground truth and prediction. The performance of the two targets, B-Score and RG, was assessed separately.

## Bacterial strains and growth conditions

Bacterial strains used in this study are listed in **S11 Table**. Unless otherwise indicated, all the strains were grown at 37˚C in Luria-Bertani (LB) media with shaking (230 rpm).

## Compound screening

Overnight cultures of the strains were back diluted to $OD_{600nm}$ of 0.036 in fresh LB medium. 100μL of cells were then arrayed onto flat-bottomed 96-well plates (Greiner Bio-One or Sarstedt, Inc.) containing 100μL of 100μM compounds, resulting in a total volume of 200μL at a final compound concentration of 50μM and final $OD_{600nm}$ of the strains 0.018. All compounds were dissolved in neat DMSO. The plates were incubated at 37˚C for 5 hours, and $OD_{600nm}$ readings were taken using BioTeK Synergy 2 plate reader. Residual Growth (RG) was calculated based on the growth ratio in the presence of compounds and growth in the DMSO control. The identity of the STL558147 was confirmed with nuclear magnetic resonance (NMR) spectroscopy. The obtained NMR spectrum was compared with those of STL558147 (provided by the supplier) and rifampicin (**S3 Fig**).

## CRISPRi Mutant construction and Rhamnose IC$_{50}$ determination

CRISPRi knockdown mutants targeting the *rpoBC* were created as previously mentioned [25]. The rhamnose concentration that inhibited 50% of mutant growth (Rha IC$_{50}$) compared to the wild-type was determined by growing the mutants in a rhamnose gradient (with trimethoprim 100μg/mL) at 37˚C with shaking (230 rpm) in 96-well format. $OD_{600nm}$ reading was taken after 20 hours of growth, and a dose-response curve was created with Graphpad PRISM version 6.0.0 (www.graphpad.com). Rha IC$_{50}$ values were calculated from the rhamnose dose-response curve using the Hill coefficient of the equation.

## Enhanced sensitivity assay

Overnight cultures of the CRISPRi mutants were diluted at 1:100 and sub-cultured with the Rha IC$_{50}$ (in LB with trimethoprim 100μg/mL) for 4h at 37˚C with shaking (230 rpm). After 4h, $OD_{600nm}$ of the cultures were adjusted to 0.01 and grown in a 96-well format containing LB broth supplemented with trimethoprim 100μg/mL, rhamnose (Rha IC$_{50}$) and various concentrations of STL558147, novobiocin, or rifampicin. The plates were incubated for 20-22h at 37˚C with shaking (230 rpm). $OD_{600nm}$ readings were taken using BioTek Synergy 2 microplate reader.

## Checkerboard assay and synergy calculation

Checkerboard assay was performed as described before [50,51]. Briefly, overnight cultures of the studied strains were back diluted to the equivalent 0.5 McFarland standard. Diluted cultures were further diluted to 1:100 in cation-adjusted Mueller Hinton Broth (CAMHB) and inoculated into 96-well plates containing a two-dimensional gradient of the STL558147 and various antibiotics, starting from half of the MICs. Synergy scores were calculated using SynergyFinder 2.0, available at https://synergyfinder.fimm.fi/ [31]. The effect of the drug combinations (synergistic or antagonistic) was calculated by comparing the observed responses against the expected response, computed using a reference model. Bliss [52], Loewe [53], highest single agent (HSA) [54] and zero interaction potency (ZIP) [55] were used as reference models to determine the degree of interaction. The drug combinations with synergy scores above 15 calculated by all models were considered synergistic; between -5 and 15 were deemed to be additive, whereas synergy scores below -15 were considered antagonistic. Synergy scores represent the mean response deviated from the reference model due to interactions between the combined drugs [14,56].

## Compound comparison between libraries

Compounds present in the FDA-approved library were filtered against the HTS dataset [14] to remove any duplicate compounds. To begin, SMILES strings for the compounds in the FDA-approved library were compared to the those in the other library. Once identical smiles strings were filtered out, Openeye Scientifics Rapid Overlay of Chemical Structures (ROCS) was utilized to calculate Tanimoto Combo (TC) scores for each molecule in the FDA-approved library to each compound in the HTS dataset. A TC score was given based on the volume and color overlap between two molecules. The color was assigned based on a molecule's hydrogen bonding, hydrophobic, aromatic, or ionic substructures. The score is a range between 0 and 2 with 2 being full color and shape overlap and 0 being no overlap.

## Supporting information

**S1 Fig. Bioactivity classes of the 100 top-ranked compounds from the FDA-approved compound library selected for experimental validation.**
(TIF)

**S2 Fig. Experimental validation of 81 top-ranked compounds from an FDA-approved compound library.** The screening identified 17 bioactive compounds with a positive predictive value (PPV) of 25.9%. The activity of growth inhibitory and non-growth inhibitory compounds are shown in red and blue, respectively. Results are the average of three independent biological replicates. Error bars indicate mean ± SD.
(TIFF)

**S3 Fig. Nuclear magnetic resonance (NMR) spectroscopy of STL558147.** (A) Comparison of STL558147 NMR spectrum with the reference spectrum provided by the supplier. The peaks in the reference spectrum are all accounted for in our NMR (highlighted by red boxes). (B) Comparison of STL558147 and rifampicin NMR spectra [57]. The differences are highlighted with red boxes.
(TIF)

**S4 Fig. Combinatorial growth inhibitory activity of STL558147 and rifampicin with ceftazidime, colistin and polymyxin B against *B*. *cenocepacia* K56-2.** Growth inhibition matrix of STL558147 (A) and rifampicin (B) in combination with ceftazidime, colistin and polymyxin B

against *B. cenocepacia* K56-2. STL558147 substantially inhibits the *B. cenocepacia* K56-2 growth (when used in combination) at a concentration ~8-fold lower than the minimum inhibitory concentration (MIC). The values in each matrix represent the mean (top) and SEM (bottom) of three independent biological replicates. The most synergistic area in each combination is highlighted with a rectangular box inside the plot. Green (negative δ-scores) indicate antagonistic interactions, and red (positive δ-scores) indicate synergistic interactions. Synergy scores >15, between -5 to 15, and < -15 were considered synergistic, additive, and antagonistic, respectively. Results are the average of at least three independent biological replicates. Synergy scores are shown as mean ± SEM. Synergy scores were calculated using SynergyFinder 2.0 [31]. (TIF)

**S5 Fig. Synergy matrix of STL558147 in combination with other antibiotics against *B. cenocepacia* K56-2.** STL558147 displayed additive interactions with rifampicin (A) and rifabutin (B), whereas exhibited antagonistic interaction with ciprofloxacin (C). The most synergistic area in each combination is highlighted with a rectangular box inside the plot. Green (negative δ-scores) indicate antagonistic interactions, and red (positive δ-scores) indicate synergistic interactions. Synergy scores >15, between -5 to 15, and < -15 were considered synergistic, additive, or antagonistic, respectively. Results are the average of at least three independent biological replicates. Synergy scores are shown as mean ± SEM. Synergy scores were calculated using SynergyFinder 2.0 [31]. (TIFF)

**S6 Fig.  Structures of PHAR261659 (A) and STL529920 (B)**. STL529920 is a stereoisomer of PHAR261659. (TIF)

**S7 Fig. Similarity comparison of the active compounds in the training/test sets with the FDA-approved compound library based on Tanimoto Combo (TC) scores.** The score is a range between 0 and 2 with 2 being full color and shape overlap and 0 being no overlap. (a) Two molecules (Clioquinol and Cetylpyridinium) in the FDA-approved library were also present in the HTS dataset set (i.e., TC score of 2). (b) Two molecules in the FDA-approved library (Oxyquinoline and Clioquinol) had a similar structure to three molecules in the HTS dataset (Chloroxine, Cloxyquin and Broxyquinoline). (TIF)

**S1 Table. Classification (Random split).** (XLSX)

**S2 Table. Classification (Scaffold split).** (XLSX)

**S3 Table. Regression (Random split).** (XLSX)

**S4 Table. Regression (Scaffold split).** (XLSX)

**S5 Table. Prediction of growth inhibitory activity in an FDA-approved compound library.** (XLSX)

**S6 Table. Growth inhibitory activity of top-ranked FDA-approved compounds against *B. cenepacia* K56-2.** (XLSX)

**S7 Table. Natural Product library.**
(XLSX)

**S8 Table. Top ranked compounds from the natural product library.**
(XLSX)

**S9 Table. Synergy score for STL558147 in combination with other clinically used antibiotics.**
(PDF)

**S10 Table. Analogs of PHAR261659 (related to Figs 3B and 5).**
(PDF)

**S11 Table. Bacterial strains and plasmids used in this work.**
(PDF)

## Acknowledgments

We thank Dr. Ayush Kumar for providing strains of ESKAPE pathogens.

## Author Contributions

**Conceptualization:** Rebecca Davis, Pingzhao Hu, Silvia T. Cardona.

**Data curation:** Chengyou Liu, Hunter Sturm, Pingzhao Hu, Silvia T. Cardona.

**Formal analysis:** Pingzhao Hu, Silvia T. Cardona.

**Funding acquisition:** Rebecca Davis, Pingzhao Hu, Silvia T. Cardona.

**Investigation:** A. S. M. Zisanur Rahman, Andrew M. Hogan, Silvia T. Cardona.

**Methodology:** A. S. M. Zisanur Rahman, Chengyou Liu, Hunter Sturm.

**Project administration:** Silvia T. Cardona.

**Resources:** Silvia T. Cardona.

**Software:** Chengyou Liu, Hunter Sturm.

**Supervision:** Rebecca Davis, Pingzhao Hu.

**Validation:** A. S. M. Zisanur Rahman.

**Visualization:** A. S. M. Zisanur Rahman.

**Writing – original draft:** A. S. M. Zisanur Rahman, Chengyou Liu, Andrew M. Hogan, Silvia T. Cardona.

**Writing – review & editing:** A. S. M. Zisanur Rahman, Hunter Sturm, Andrew M. Hogan, Rebecca Davis, Pingzhao Hu, Silvia T. Cardona.

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
