## [Decision Letter · Decision Letter 0]

18 Jul 2022

Dear Dr. Cardona,

Thank you very much for submitting your manuscript "A Machine Learning Model trained on a High-Throughput Antibacterial Screen Increases the Hit Rate of Drug Discovery" for consideration at PLOS Computational Biology.

As with all papers reviewed by the journal, your manuscript was reviewed by members of the editorial board and by several independent reviewers. In light of the reviews (below this email), we would like to invite the resubmission of a significantly-revised version that takes into account the reviewers' comments.

The reviewers made specific suggestions for clarifications and raised additional questions. If you decide to submit a revised version, these points should be addressed carefully. Please respond especially to the two major issues raised by Reviewer 3. Also, please make sure that the source code is made available according to the guidelines of the journal.

We cannot make any decision about publication until we have seen the revised manuscript and your response to the reviewers' comments. Your revised manuscript is also likely to be sent to reviewers for further evaluation.

Sincerely,

Tobias Bollenbach

Associate Editor

PLOS Computational Biology

Feilim Mac Gabhann

Editor-in-Chief

PLOS Computational Biology

The reviewers made a few suggestions for clarifications and raised additional questions. If you decide to submit a revised version, these points should be addressed carefully. Please respond especially to the two major issues raised by Reviewer 3. Also, please make sure that the source code is made available according to the guidelines of the journal.

Reviewer's Responses to Questions

**Comments to the Authors:**

Reviewer #1: This is an excellent paper and for once I have absolutely no comments. It is well written, nicely presented and will become a reference in the field. They also acknowledge previous publications appropriately.

Reviewer #2: This is an interesting study. The authors built a machine learning model with antibacterial dataset, which was further used for high-throughtput screening. The hit rate proved to be increased. I apprecriate using Bayesian optimization for hyperparameter optimization. The predictions were further evaluated with experiments. The manuscript is well-written and I would suggest to accept this manuscript after minior revision.

Here are some questions that I have.

1. On page 18, there are some evaluation metrices reported. Can you also add the results of Matthews correlation coefficient (MCC). The advantages of MCC is reported in https://bmcgenomics.biomedcentral.com/articles/10.1186/s12864-019-6413-7.

2. All the source codes were not listed in the manuscript and it would be very helpful if they can provided to support open science.

Reviewer #3: In this manuscript, the authors trained an ML model with data from a high throughput screening experiment, and used this ML model to predict compounds with antibacterial activity in the library of FDA-approved compounds and natural products. Then, some compounds with growth inhibitory activity against several Gram-negative bacteria were identified by wet experiments. This manuscript combines experiments and computation well and has the potential to be accepted. However, the reviewer has some concerns and suggestions.

Major Comments:

(1) The authors claimed many times in the article that their approach increases the hit rate of drug discovery by 12-fold at least. However, the hit rate of the virtual screen has a great relationship with the experimental system and the definition of hits. It is not appropriate to quantitatively compare the hit rate in this study with that of the hit rate from conventional whole-cell-based high-throughput screens. I recommend that the authors compare the hit rate in this study with the hit rate from their previously performed HTS (the training dataset).

(2) Most of the compounds screened by the ML model from the FDA-approved compound library are known antibiotics. Are there very similar molecules to those hit compounds in the training set? Please provide the maximum similarity between these hit compounds and the compounds in the training set.

Minor Comments:

(1) The compound structures in Fig 3 and Fig 6 are not clear enough.

(2) Is the training dataset in this study available?

**Have the authors made all data and (if applicable) computational code underlying the findings in their manuscript fully available?**

Reviewer #1: Yes

Reviewer #2: **No: **Source code would be useful.

Reviewer #3: None

PLOS authors have the option to publish the peer review history of their article (what does this mean?). If published, this will include your full peer review and any attached files.

Reviewer #1: No

Reviewer #2: No

Reviewer #3: No
---

## [Decision Letter · Decision Letter 1]

26 Sep 2022

Dear Dr. Cardona,

We are pleased to inform you that your manuscript 'A Machine Learning Model trained on a High-Throughput Antibacterial Screen Increases the Hit Rate of Drug Discovery' has been provisionally accepted for publication in PLOS Computational Biology.

In addition, ensure that the github repository is active, correctly listed, and publicly available.

Best regards,

Tobias Bollenbach

Academic Editor

PLOS Computational Biology

Feilim Mac Gabhann

Editor-in-Chief

PLOS Computational Biology

Both reviewers point out that the provided github link to the code does not work. Please make sure that the source code is made available according to the guidelines of the journal.

Reviewer's Responses to Questions

**Comments to the Authors:**

Reviewer #2: I am happy with all the changes. The only concern that I have is the GitHub repo is not accessible to the public, https://github.com/cardonalab/Prediction-of-ATB-Activity. The manuscript can be accepted after fixing this.

Reviewer #3: No furthe comments. The code link is not valid.

https://github.com/cardonalab/Prediction-of-ATB-Activity

The other issues have been addressed, I'll leave this to the editor to decide if it's acceptable.

**Have the authors made all data and (if applicable) computational code underlying the findings in their manuscript fully available?**

Reviewer #2: **No: **I would guess that the GitHub repo is a private one.

Reviewer #3: **No: **The code link is not valid.

https://github.com/cardonalab/Prediction-of-ATB-Activity

The other issues have been addressed, I'll leave this to the editor to decide if it's acceptable.

PLOS authors have the option to publish the peer review history of their article (what does this mean?). If published, this will include your full peer review and any attached files.

Reviewer #2: No

Reviewer #3: **Yes: **Mingyue Zheng

---

## [Editor Report · Acceptance letter]

7 Oct 2022

PCOMPBIOL-D-22-00724R1 

A Machine Learning Model trained on a High-Throughput Antibacterial Screen Increases the Hit Rate of Drug Discovery

Dear Dr Cardona,

I am pleased to inform you that your manuscript has been formally accepted for publication in PLOS Computational Biology. Your manuscript is now with our production department and you will be notified of the publication date in due course.

With kind regards,

Anita Estes
